# All-dielectric chiral-field-enhanced Raman optical activity

Ting-Hui Xiao [1,2✉], Zhenzhou Cheng[1,3], Zhenyi Luo[1], Akihiro Isozaki [1], Kotaro Hiramatsu [1,4], Tamitake Itoh[5], Masahiro Nomura [6], Satoshi Iwamoto [6,7] & Keisuke Goda [1,2,8,9✉]

Raman optical activity (ROA) is effective for studying the conformational structure and behavior of chiral molecules in aqueous solutions and is advantageous over X-ray crystallography and nuclear magnetic resonance spectroscopy in sample preparation and cost performance. However, ROA signals are inherently minuscule; 3–5 orders of magnitude weaker than spontaneous Raman scattering due to the weak chiral light–matter interaction. Localized surface plasmon resonance on metallic nanoparticles has been employed to enhance ROA signals, but suffers from detrimental spectral artifacts due to its photothermal heat generation and inability to efficiently transfer and enhance optical chirality from the far field to the near field. Here we demonstrate all-dielectric chiral-field-enhanced ROA by devising a silicon nanodisk array and exploiting its dark mode to overcome these limitations. Specifically, we use it with pairs of chemical and biological enantiomers to show >100x enhanced chiral light–molecule interaction with negligible artifacts for ROA measurements.

[1] Department of Chemistry, The University of Tokyo, Tokyo, Japan. [2] Institute for Quantum Life Science, National Institute for Quantum and Radiological Science and Technology, Chiba, Japan. [3] School of Precision Instruments and Opto-electronics Engineering, Tianjin University, Tianjin, China. [4] PRESTO, Japan Science and Technology Agency, Saitama, Japan. [5] Health and Medical Research Institute, National Institute of Advanced Industrial Science and Technology, Takamatsu, Japan. [6] Institute of Industrial Science, The University of Tokyo, Tokyo, Japan. [7] Research Center for Advanced Science and Technology, The University of Tokyo, Tokyo, Japan. [8] Institute of Technological Sciences, Wuhan University, Wuhan, Hubei, PR China. [9] Department of Bioengineering, University of California, Los Angeles, CA, USA. ✉email: xiaoth@chem.s.u-tokyo.ac.jp; goda@chem.s.u-tokyo.ac.jp

Raman optical activity (ROA) is an effective tool for identifying the absolute configuration of chiral molecules in aqueous solutions by measuring their small Raman scattering intensity difference between right-circularly polarized (RCP) and left-circularly polarized (LCP) incident light[1–5]. Since its first demonstration of ROA in the early 1970s[4], ROA has been proven useful in diverse fields including biochemistry[5–7], stereochemistry[8,9], analytical chemistry[10,11], structural virology[12], and pharmaceutical science[13] by virtue of its ability to determine the conformational structure and behavior of chiral biomolecules in aqueous solutions such as proteins, nucleic acids, carbohydrates, viruses, and biopolymers. As a structure-sensitive spectroscopic method that directly probes the handedness of molecular vibrations via Raman scattering[1], ROA is advantageous in sample preparation and cost performance over the commonly used non-optical methods including X-ray crystallography[14,15] and nuclear magnetic resonance (NMR) spectroscopy[16]. Furthermore, ROA is accessible to intrinsically disordered or natively unfolded proteins (without a compact tertiary fold) specified by genomes, while X-ray crystallography and NMR spectroscopy fail to probe their absolute configurations due to extreme difficulties in crystalizing molecules[17].

Unfortunately, ROA is extremely weak; ROA signals are 3–5 orders of magnitude weaker than Raman scattering signals since the optical activity of natural chiral molecules is minuscule[18,19]. This property not only makes the ROA signals easily obscured by spectral artifacts[20,21], but also results in a single signal acquisition time up to tens of hours (sometimes days)[22], seriously limiting ROA's practical utility. As a result, ROA is much less popular among users than X-ray crystallography and NMR spectroscopy. To overcome this limitation, efforts have been made to increase the ROA signal level via surface-enhanced ROA in which the ROA signal is enhanced by exciting localized surface plasmon resonance (LSPR) on metallic nanoparticles[18]. However, the realization of surface-enhanced ROA with few spectral artifacts and high reproducibility remains a significant challenge for a few critical reasons. First, LSPR is not able to efficiently transfer and enhance the optical chirality of incident circularly polarized light from the far field to the near field for ROA excitation[18,20]. Second, the random motion and disordered arrangement of the metallic nanoparticles suspended in aqueous solutions seriously change their polarization dependence during the long integration time and introduce unamendable artifacts into the surface-enhanced ROA signal[20]. Third, the metallic nanoparticles generate large photothermal heat, which aggravates the instability of the ROA signal and destroys the conformation of chiral molecules, especially when they are biological[23].

In this article, we theoretically proposed and experimentally demonstrated all-dielectric[24–28] (i.e., metal-free) chiral-field-enhanced ROA by tailoring a chiral field[29,30] in an array of silicon nanodisks to significantly increase the interaction between incident circularly polarized light and chiral molecules via exploiting a dark mode in the silicon nanodisk array, thereby overcoming all the limitations of LSPR-based ROA. Here, the dark mode is a combination of electric and toroidal dipoles that results in their partial destructive interference in the far field[31,32]. Specifically, we designed and fabricated an optically isotropic silicon nanodisk array on a chip, which enabled precise tailoring of optical chirality in the near-field region during the signal acquisition time and avoided introducing additional artifacts into enhanced ROA measurements. In other words, the dark mode in the silicon nanodisk array was excited to generate an optimized chiral field, which not only enabled optical chirality enhancement and its efficient transfer from the far field to the near field, but also allowed a substantial level of spatial overlap between the enhanced electric field and optical chirality in the silicon nanodisk array. In addition to the physical advantages of the silicon nanodisk array, its fabrication process is fully CMOS-compatible and applicable for integration with other on-chip devices as well as high-volume production for chiral sensing. To show the practical utility of our method, we conducted ROA measurements of pairs of chemical and biological enantiomers, namely (±)-alpha-pinene and (±)-tartaric acid, with negligible artifacts in a two-phase virtual-enantiomer ROA setup, which indicates an ROA signal enhancement factor of ~$10^2$ in the near-field region. Our method is expected to open an efficient, cost-effective, reliable way for absolute conformational analysis of trace chiral molecules which cannot be analyzed by X-ray crystallography and NMR spectroscopy and hence holds promise in diverse fields such as analytical chemistry, structural virology, and pharmaceutical science.

## Results and discussion

**Design of the silicon nanodisk array.** The concept of chiral-field-enhanced ROA by the silicon nanodisk array is schematically illustrated in Fig. 1a. A beam of circularly polarized light is normally incident on natural chiral molecules on the silicon nanodisk array. An ROA signal enhanced by the silicon nanodisk array is collected to analyze the absolute conformational structure of the chiral molecules. The silicon nanodisk array which consists of nanodisks with radius $r$ and height $h$ is designed to be packed in a square lattice with period $p$. Such a geometric configuration with $C_4$ rotational symmetry enables an isotropic response of the silicon nanodisk array to incident light (Supplementary Note 1), which prevents introducing birefringence artifacts into the chiral-field-enhanced ROA signal. Figure 1b shows a simulated spectrum of the reflection of normally incident light from the silicon nanodisk array as a function of period $p$ with a constant $r$ of 90 nm and a constant h of 180 nm. A dark mode is present in the low-reflectivity region in the figure. The excitation wavelength in the simulation is selected at a wavelength of 532 nm by taking both Raman scattering efficiency and optical losses of the silicon nanodisks into account[33]. At this excitation wavelength, the chiral field in the near-field region of the silicon nanodisk array is tailored by tuning the excited mode by varying the geometric parameters of the nanodisks. Figure 1c, d shows the average electric-field magnitude and the average optical activity in the near-field region of the silicon nanodisk array as a function of $r$ and $h$ with a constant gap size ($p − 2r$) of 30 nm. Different resonant modes can be selectively excited by using the different geometric parameters of the nanodisks, which are indicated by the high-electric-field regions in Fig. 1c. Among these resonant modes, a dark mode[31] exhibits not only a high electric-field magnitude for Raman enhancement, but also a high optical chirality in the near field for ROA enhancement as shown in Fig. 1d. This indicates the ability of the dark mode to efficiently transfer and enhance the optical chirality of the incident circularly polarized light from the far field to the near field. The magnetic-field and electric-field magnitude distributions of the dark mode in a unit cell of the silicon nanodisk array with a nanodisk radius $r$ of 90 nm and a nanodisk height $h$ of 180 nm are simulated and shown in Fig. 1e, f, respectively. The results verify the excitation of the dark mode as well as its high electric-field enhancement (Supplementary Note 2). With the excitation of this dark mode, the optical chirality distributions in the unit cell with LCP and RCP light excitation are shown in Fig. 1g. The figure indicates that the generation of chiral fields with oppositely enhanced optical chirality is ideal for ROA enhancement. In other words, both the electric field and optical chirality enhancements contribute to the ROA enhancement, but with distinct physical mechanisms. Specifically, the electric-field enhancement increases

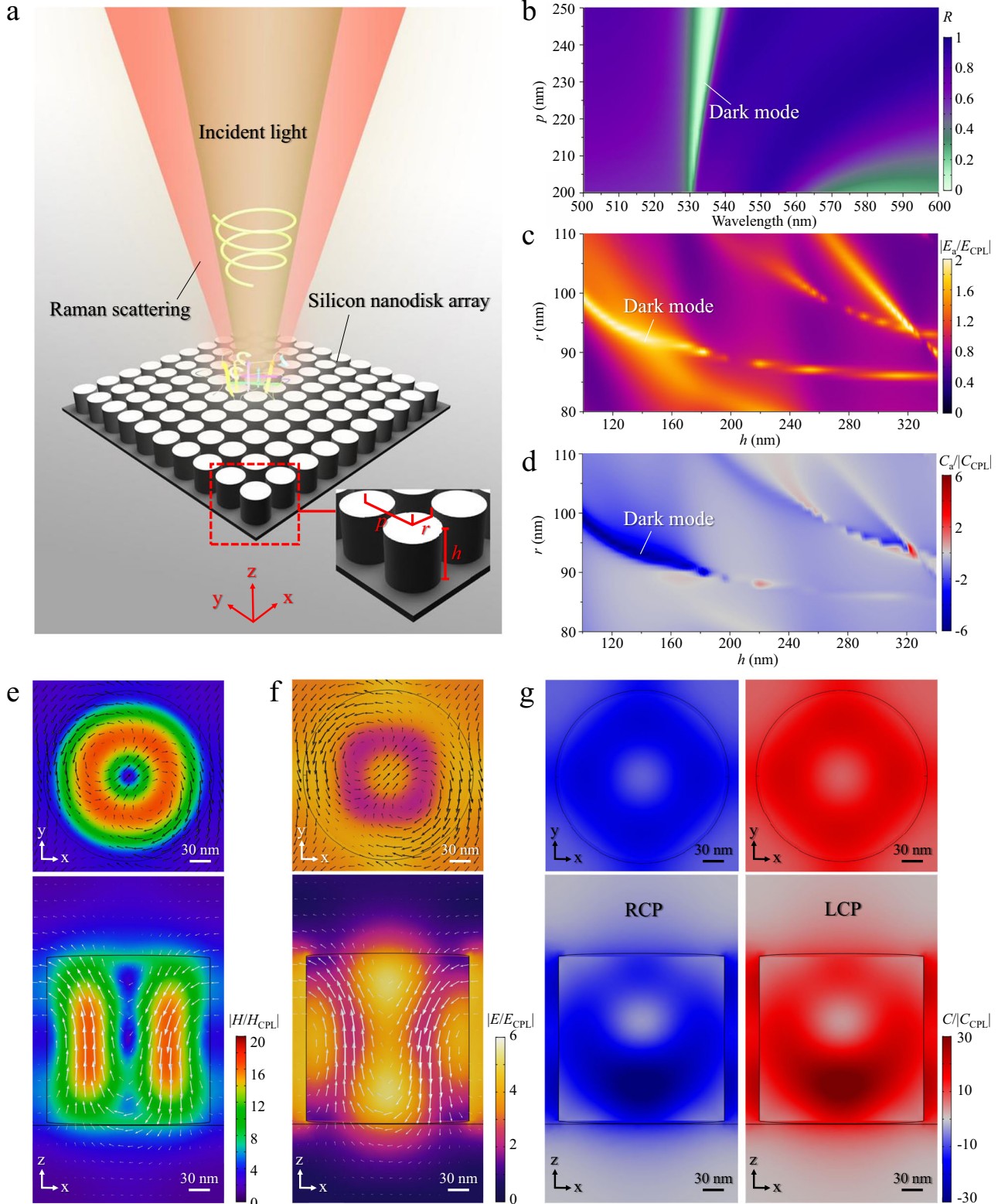

the ROA signal by enhancing the Raman scattering signal from chiral molecules, whereas the optical-chirality enhancement enables an enlarged chiral part of the Raman excitation rate of the chiral molecules. Moreover, the large spatial overlap between the electric-field and optical-chirality enhancements indicates that the dark mode is promising for efficient ROA enhancement as the spatial overlap determines the efficiency of optical-chirality

enhancement in the near field (Supplementary Note 3 and Supplementary Fig. 1).

**Fabrication and evaluation of the silicon nanodisk array**. Based on the above theoretical design, we experimentally fabricated and characterized the silicon nanodisk array on a silicon-on-insulator

**Fig. 1 Principle of chiral-field-enhanced ROA using the silicon nanodisk array. a** Schematic of the silicon nanodisk array. **b** Reflection spectrum of the silicon nanodisk array as a function of period $p$. **c** Electric-field magnitude in the near-field region on the silicon nanodisk array as a function of the dimensions of the nanodisks, normalized by the electric-field magnitude of incident circularly polarized light (CPL) in the far field, at the wavelength of 532 nm. **d** Average optical activity (RCP excitation) in the near-field region on the silicon nanodisk array as a function of the dimensions of the nanodisks, normalized by the optical chirality of incident CPL in the far field, at the wavelength of 532 nm. **e** Top and side views of the magnetic-field magnitude distribution in a unit cell of the nanodisk array, normalized by the magnetic-field magnitude of incident CPL in the far field, at the wavelength of 532 nm. The white arrows indicate the direction of the magnetic field at each point. **f** Top and side views of the electric-field magnitude distribution in a unit cell of the nanodisk array, normalized by the electric-field magnitude of incident CPL in the far field, at the wavelength of 532 nm. The black arrows indicate the direction of the electric field at each point, while the white arrows indicate the direction of the magnetic field at each point. **g** Top and side views of the optical chirality distribution in a unit cell of the nanodisk array, normalized by the optical chirality of incident CPL, at the wavelength of 532 nm.

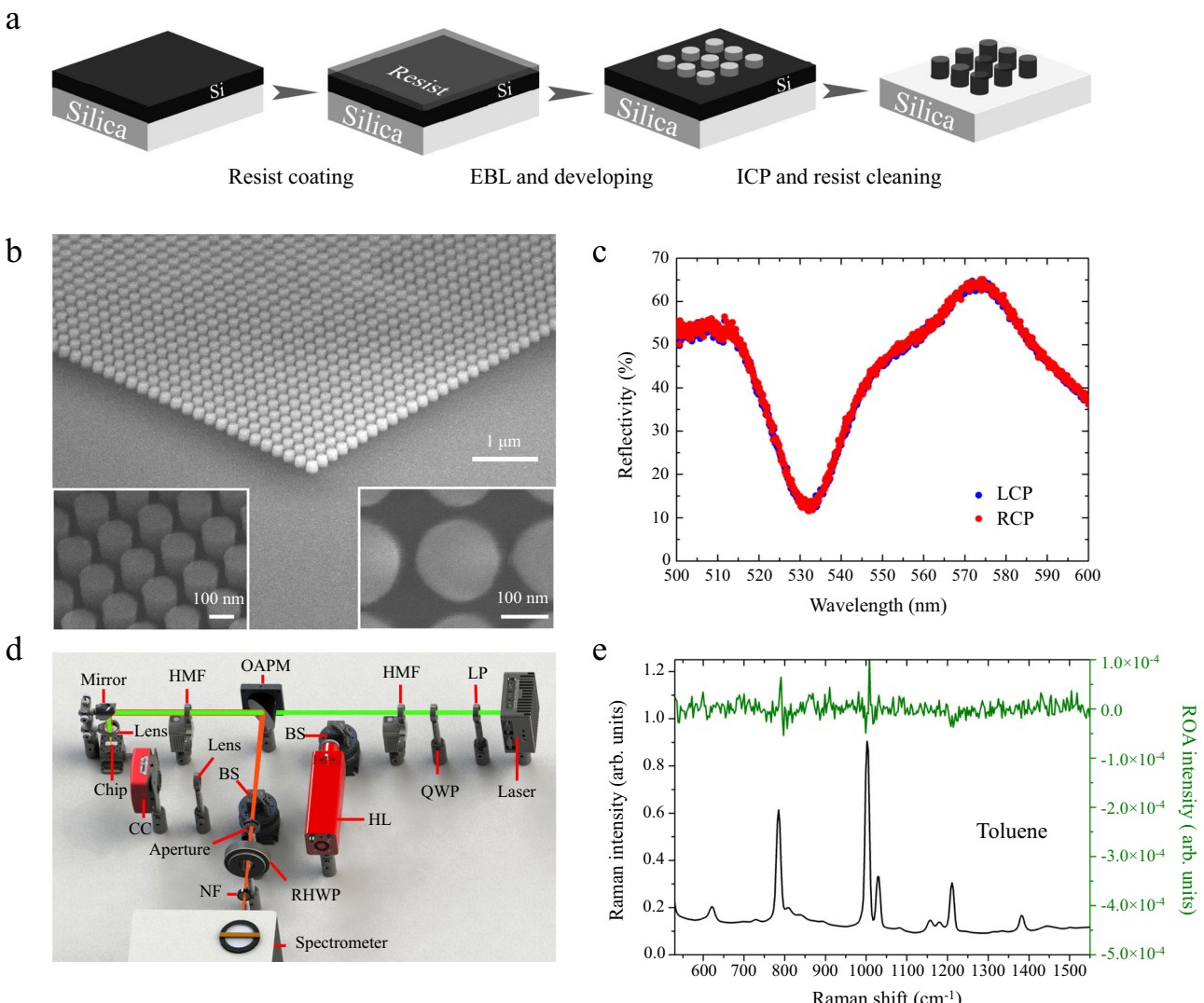

**Fig. 2 Experimental chiral-field-enhanced ROA system. a** Fabrication of the silicon nanodisk array. EBL: electron beam lithography. ICP: inductively coupled plasma etching. **b** SEM images of the fabricated silicon nanodisk array. The insets show enlarged SEM images of the nanodisk array. **c** Reflection spectra of the silicon nanodisk array excited by incident LCP and RCP light. **d** Two-phase virtual-enantiomer ROA setup. LP linear polarizer, QWP quarter-wave plate, HMF half-wave plate in a motorized flipper, BS beam splitter, HL halogen lamp, OAPM off-axis parabolic mirror, RHWP rotating half-wave plate, NF notch filter, CC CMOS camera. **e** Chiral-field-enhanced Raman and ROA spectra of toluene (an achiral sample).

wafer (Fig. 2a and Supplementary Note 4). Specifically, we used thin-film crystalline silicon instead of amorphous or poly-crystalline silicon to significantly reduce optical losses in the silicon nanodisk array at a wavelength of 532 nm[33]. A scanning electron microscope (SEM) image of our fabricated square-lattice silicon nanodisk array is shown in Fig. 2b. It shows that the geometric parameters of the fabricated nanodisk array fit well

with the theoretical design. After the fabrication, we characterized the silicon nanodisk array by measuring its reflection spectra with indent LCP and RCP light in a home-made reflection spectro-scopy setup (Supplementary Note 5 and Supplementary Fig. 2) as shown in Fig. 2c. The measured two reflection spectra are in good agreement, indicating the achiral property of the silicon nanodisk array. The lineshapes of the reflection spectra agree well with the

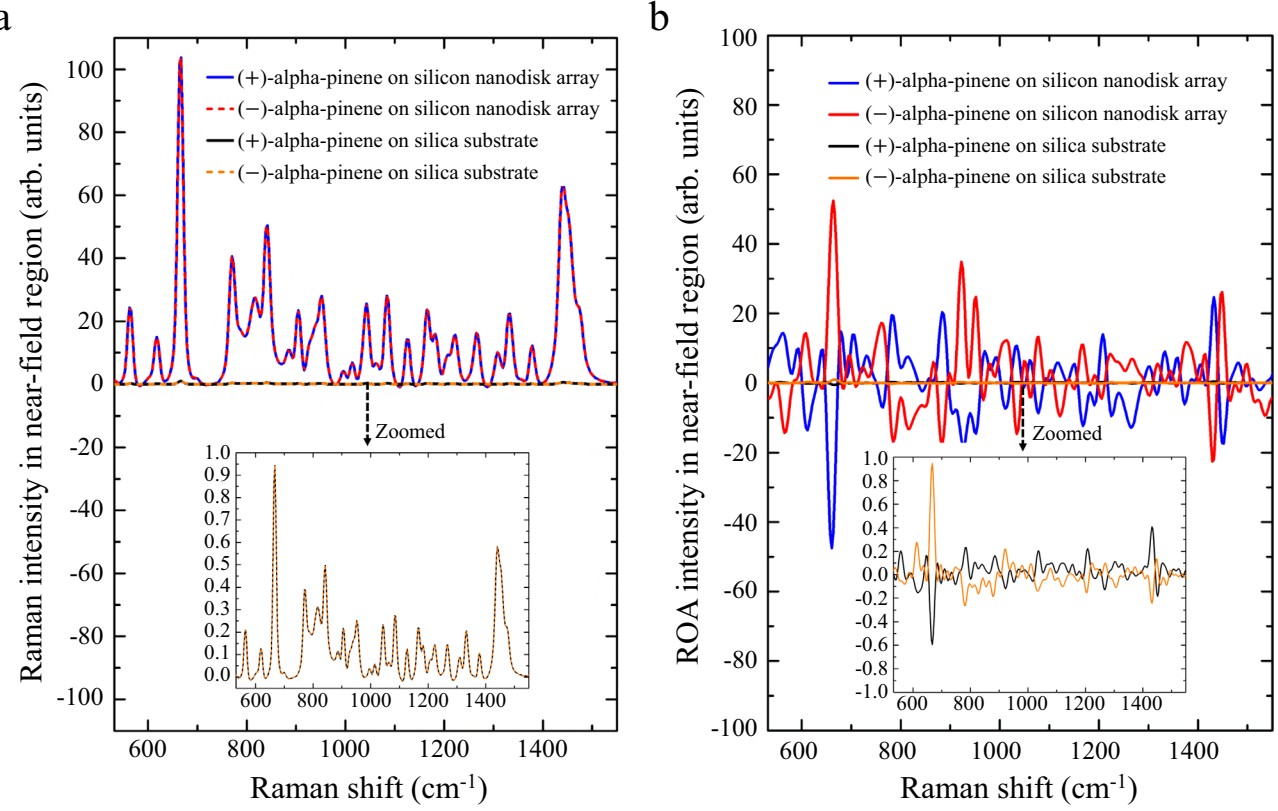

**Fig. 3 Raman and ROA spectra of (±)-alpha-pinene (a pair of enantiomers) with and without the chiral-field enhancement by the silicon nanodisk array. a** Raman spectra of (±)-alpha-pinene on the silica substrate and the silicon nanodisk array. The Raman intensity is ~100x enhanced by the silicon nanodisk array. **b** ROA spectra of (±)-alpha-pinene on the silica substrate and the silicon nanodisk array. The ROA intensity is ~100x enhanced by the silicon nanodisk array.

simulations, where a reflection valley appears at the wavelength of 532 nm. To check potential artifacts resulting from the silicon nanodisk array, we used a home-made two-phase virtual-enantiomer ROA setup[34] (Fig. 2d and Supplementary Note 6) to obtain the enhanced Raman and ROA spectra of toluene (an achiral sample). As shown in Fig. 2e, the measured ROA of toluene is comparable to the shot noise level, indicating an artifact-suppressed condition of our chiral-field-enhanced ROA system.

**Performance of the chiral-field-enhanced ROA**. To demonstrate the performance of our silicon nanodisk array for chiral-field-enhanced ROA, we measured ROA spectra of alpha-pinene, a well-known chiral molecule. With an incident laser power of 800 mW and a total exposure time of 5 h, we obtained the Raman spectra of a trace amount of (±)-alpha-pinene (25-μm-thick layers, effective volume: ~0.1 pl) on the silica substrate and silicon nanodisk array under the same experimental conditions (Fig. 3a). Here, the Raman spectra were normalized by the maximum Raman signal intensity value obtained on the silica substrate. It is evident from the figure that the measured Raman spectra of this pair of enantiomers agree well. As each of the Raman spectra is the sum of the Raman spectra of the molecule excited by incident LCP and RCP light, their agreement verifies the optically achiral properties of both the silica substrate and the silicon nanodisk array. Moreover, compared with the Raman spectra measured on the silica substrate, a Raman signal enhancement with an average enhancement factor of ~10² in the near-field region, which was obtained by removing the far-field contribution, was evident on at the silicon nanodisk array (Supplementary Note 7

and Supplementary Figs. 3 and 4). Correspondingly, the measured ROA spectra of this pair of enantiomers on the silica substrate and silicon nanodisk array are shown in Fig. 3b in which the ROA spectra were normalized by the maximum ROA signal intensity value obtained on the silica substrate. The mirror-symmetric ROA spectra indicate an artifact-suppressed condition of the chiral-field-enhanced ROA system, verifying the stability and reliability of the silicon nanodisk array for ROA enhancements. Similar to the Raman spectra, an ROA signal enhancement with an average enhancement factor of ~10² in the near-field region, which was obtained by removing the far-field contribution, was evident for both (±)-alpha-pinene (Supplementary Note 7 and Supplementary Figs. 3 and 4). Moreover, the similar enhancement factors of the Raman and ROA signals show no appreciable difference in circular intensity difference (CID) between the far-field and near-field ROA measurements, meaning that the dissymmetric factor of the incident circularly polarized light, which is the intensity-normalized optical chirality that reflects the chiral structure of the optical field, is unchanged in the near field with respect to the far field even when the light is transferred and confined to the nanoscale via the silicon nanodisk array. This agrees well with our theoretical prediction in which CID ∝ *g*, where *g* is the dissymmetric factor (Supplementary Note 8 and Supplementary Fig. 6).

To demonstrate the compatibility and applicability of the silicon nanodisk array to biological molecules for which ROA is the most effective, we conducted ROA measurements of tartaric acid, a chiral biomolecule typically found in plants[35] and widely used in food and drugs. With an incident laser power of 900 mW and a total exposure time of 6 h, we obtained the Raman and

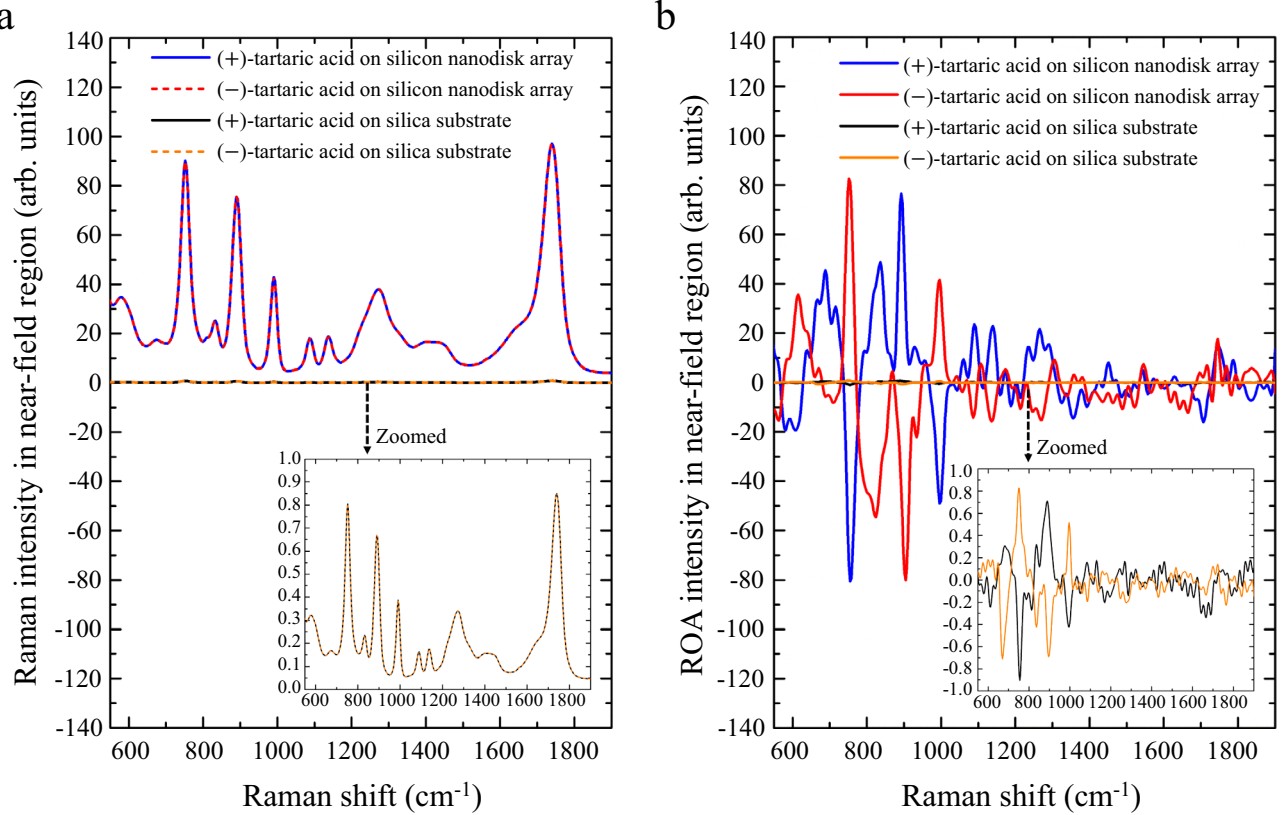

**Fig. 4 Raman and ROA spectra of (±)-tartaric acid (a pair of biological enantiomers) with and without the chiral-field enhancement by the silicon nanodisk array. a** Raman spectra of (±)-tartaric acid on the silica substrate and the silicon nanodisk array. The Raman intensity is ~100x enhanced by the silicon nanodisk array. **b** ROA spectra of (±)-tartaric acid on the silica substrate and the silicon nanodisk array. The ROA intensity is ~100x enhanced by the silicon nanodisk array.

ROA spectra of a trace amount of (±)-tartaric acid (25-μm-thick solution layers, effective volume: ~0.1 pl, concentration: 5 M) on the silica substrate and silicon nanodisk array under the same experimental conditions. Compared with the Raman spectra measured on the silica substrate, enhanced Raman spectra of both (±)-tartaric acid were measured on the silicon nanodisk array as shown in Fig. 4a in which the Raman spectra were normalized by the maximum Raman signal intensity value obtained on the silica substrate. Similar to the Raman spectra of (±)-alpha-pinene, an average Raman signal enhancement factor of ~$10^2$ in the near-field region was also experimentally obtained for (±)-tartaric acid (Supplementary Note 7 and Supplementary Fig. 5). Moreover, the characteristic Raman peaks of tartaric acid measured on the nanodisk array are consistent with those measured on the silica substrate, indicating the excellent biocompatibility and reliability of the silicon nanodisk array for enhanced Raman spectroscopy. More importantly, enhanced ROA signals were also observed on the silicon nanodisk array with an average enhancement factor of ~$10^2$ in the near-field region as shown in Fig. 4b in which the ROA spectra were normalized by the maximum ROA signal intensity value obtained on the silica substrate. The measured ROA characteristic peaks of this pair of biological enantiomers agree well with those measured on the silica substrate, indicating the excellent biocompatibility and reliability of the silicon nanodisk array for enhanced ROA spectroscopy by virtue of its efficient transfer and enhancement of optical chirality, reliable on-chip configuration, and low photothermal heat generation (Supplementary Note 9 and Supplementary Fig. 7).

## Data availability

The source data supporting the findings of this study are available at https://doi.org/10.5281/zenodo.3994000 and are also available from the corresponding authors upon reasonable request.

## Code availability

All codes used for analysis of this study are available from the corresponding authors upon reasonable request.

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

## Acknowledgements

This research was supported by MEXT Quantum Leap Flagship Program (JPMXS0120330644), JSPS KAKENHI (JP18K13798, JP20K14785), JSPS Core-to-Core Program, Murata Science Foundation, White Rock Foundation, and University of Tokyo GAP Fund.

## Author contributions

T.-H.X. and Z.C. conceived the study. T.-H.X. performed the theoretical calculations and experiments and analyzed the data. Z.L., A.I., K.H., T.I., M.N., and S.I. provided support for the theoretical calculations and experiments. Z.C. and K.G. supervised the study. T.-H.X., Z.C., and K.G. wrote the manuscript. All authors discussed the results and contributed to the manuscript.

## Competing interests

The authors declare no competing interests.
