## [Peer Review File · Nature Communications]

REVIEWER COMMENTS

Reviewer #1 (Remarks to the Author):

This paper represents a significant advance in ROA measurement methodology. It is shown by the use of a suitably constructed silicon nanodisk array that the Raman and ROA of molecules located in the chirally-enhanced near-field region of the array are enhanced approximately by a factor of 100. The symmetry of the nanodisk array gives rise to an anapole toroidal field which theoretically suppresses linear polarization bias of the sample that in turn allows transfer of RCP and LCP light from the far field to the enhanced near field without the introduction of ROA artifacts (however see below) and a preservation of the ratio of ROA to Raman intensities, the so-called circular intensity difference (CID). The demonstration of near-field Raman/ROA enhancement of the nanodisk array is made by comparison to the Raman/ROA far-field intensities obtained using a silicon disk of the same dimensions by lacking the array. This achievement is significant because it addresses in a fundamental way a general disadvantage of ROA, namely the small ratio of ROA to Raman intensities and the need to use high concentrations, relative high incident laser powers and long measurement times to achieve ROA spectra with good signal quality.

The paper is carefully presented with demonstrations, including critically the SI material, of several key steps including 1) the symmetry properties of the array, 2) the anapole nature of the chiral enhancing near field, 3) the effect of this field on the induced electric dipole and induced magnetic dipole moments, including the optical chirality of the incident light that in this case, contributes to the generation of the observed enhanced Raman and ROA, 4) the Raman and ROA enhancement factors for the nanodisk array and 5) the dissymmetry factor achieved by the array showing the preservation of the chiral light environment from the far field to the near field and the preservation of the CID of the ROA.

Despite this impressive achievement for the enhanced measurement of Raman and ROA, the paper exhibits several shortcomings that need to be addressed before the paper is published.

Fig 1a shows incident CP laser radiation along the vertical z-axis with Raman scattering emerging apparently in any direction in the xy-plane just above the nanodisk array, i.e. 90-degree ICP-ROA scattering, yet the instrument layout in Fig. 2d shows both the incident and 180-degree backscattering ICP-ROA occurring along the vertical z-axis and only subsequently reflected into the horizontal direction by a mirror. This needs to be corrected or clarified.

There are several issues with Fig. 3 containing chirally-enhanced near field and expanded far field Raman and ROA of both enantiomers of alpha-pinene that need comments by the authors. The Raman and ROA of neat alpha-pinene are extremely well-known in the literature and used extensively to calibrate standard ROA spectrometers. The accuracy and artifact level of the ROA can be assessed for example by comparison to the paper by W. Hug, *Appl. Spectrosc.* 57, 1-13 (2003) where the concept of virtual enantiomer is presented for the first time, and upon which the instrument design is based but only a secondary reference 29 is provided. In Figs. 9 and 10 of the Hug paper high-quality conventional Raman and ROA spectra of both enantiomers of alpha pinene are displayed. Compared to the corresponding Raman and ROA spectra in Fig. 3 of the present paper allows one to verify all the features of the Raman spectra and roughly verify the signs and magnitudes of the ROA bands. The problem with the ROA comparison is the relative low signal to noise ratio and the crowding, almost overlapping, of the ROA bands compared to the corresponding Raman bands. Such crowding does not occur in conventional ROA spectra as in, for example, the Hug paper. The Hug spectra have signal to

noise ratios approaching 100 and much higher apparent spectral resolution, 33 minutes of collection time and 420 mW incident laser power using a 35 microliter cell. The ROA spectra in Fig. 3 here have signal to noise ratio closer to only 10, 300 minutes of collection time (5h), 800 mW of laser power, and an effective volume of 0.1 picoliters. The only favorable aspect of this comparison is the effective volume of 0.1 pL that arises from the 100 fold intensity enhancement, but beyond that there is no demonstrated advantage of the superchiral ROA enhancement. The authors should be more forthcoming about the quality of the ROA spectra they present versus the quality available from existing conventional ROA methodology.

A second issue the authors need to address is their statement on line 140 of the "excellent artifact-free condition of the ROA spectra in Fig. 3. This is not correct. The most difficult problem in ROA measurement is to avoid artifacts in highly polarized Raman bands. Alpha pinene has two such bands, one at 671 cm⁻¹ and the other at 1663 cm⁻¹. The latter is not shown in the Raman and ROA spectra in Figs. 3a and 3b, respectively; however the very strong Raman band at 671 cm⁻¹ is accompanied by a very strong, presumably artifact, ROA band, the strongest in the ROA spectrum by a factor of ~2. According to the ROA literature in general and the Hug paper in particular instead of a strong ROA band 671 cm⁻¹ there should be one of the smallest ROA bands, smaller than it's neighboring band near 600 cm⁻¹. Most likely there would also be a strong ROA artifact band at 1663 cm⁻¹ if this were shown. The authors need to confront this artifact problem which calls into question the artifact-free contention of these new ROA spectra.

Third, the authors need to be more clear about the comparison of superchiral enhanced spectra displayed in Fig. 3, versus the reference non-enhanced spectra shown as zoomed enlarged spectra by a factor of approximately 100. As shown in the SI, the superchiral enhanced spectra are not direct measurements but rather spectral subtractions to remove the larger contribution from the superimposed far field spectra. This fact explains why the superchiral enhanced ROA spectra have noise levels similar to the non-enhanced spectra. The authors need to make these facts more clear and address the question of whether the superchiral near-field Raman/ROA can be seen directly without interference from the far field contributions.

In summary, the authors have demonstrated the ability to isolate and measure superchiral enhanced Raman and ROA spectra that are 100-times enhanced by the use of nanodisk array. This is a conceptual and experimentally-realized accomplishment of significant interest. There is, nevertheless, still an unresolved problem of a very large artifact ROA band at 671 cm⁻¹ that should be at least an order of magnitude smaller. Although the superchiral enhanced ROA emerges from a tiny volume of less than 1 pL, because of the presence of larger far-field Raman and ROA spectral contributions, there is still no noise improvement even though there is a 100-fold enhancement due to the superchiral fields provided by the nanodisk array. Thus, there are several issues that need to be resolved or made more clear before the paper can be accepted for publication.

Reviewer #2 (Remarks to the Author):

Dear Editors and co-authors,

I read the manuscript by Ting-Hui Xiao et.al. with great care. It presents a very promising study on the enhancement of the ROA of chiral molecules mediated by all-dielectric metasurfaces. The experimental results are very promising, and the technological implications of the technique are of high relevance. Nevertheless, I encountered several drawbacks in the analysis of the results of the article. I think that the article could deserve its publication in Nature Communications, but not in this

form. I would recommend its acceptance if the authors would address the following points convincingly:

- An anapole ideally does not radiate to the far-field. Thus, one cannot couple to it with far-field excitations either. Although the authors mention that the cancelation of toroidal and magnetic dipolar modes in their system is not ideal, they should develop and clarify this point in the main text. Also, this far from perfect cancelation, makes me think if this "mode" deserves to be called an anapolar mode. This criticism points more to a linguistic argument, it does not preclude the experimental results from being valid. Nevertheless, I would suggest the authors to rephrase a bit the words used in their theoretical interpretation. I would not refer to these resonances as anapolar modes.

- As the authors mention in the manuscript, Silicon is a lossy material In the frequency of their excitation (532nm). In the introduction, they say that heating can perturb the ROA signal of chiral molecules. Although apparently this is not a serious issue in their article (the features of the ROA spectra in their metasurface are similar to that obtained in silica), I miss an analysis of the heating in their system in comparison with other previously proposed plasmonic systems to quantify the adequateness of their experimental platform.

- I have serious doubts about the theoretical interpretation of the ROA enhancement they develop in the supplementary information, which I consider very simplistic and incomplete.

In simple words, Surface Enhanced Raman Spectroscopy (SERS), the increase in the Raman signal encompasses two different sources of enhancement. On the one hand, the molecules under study are more efficiently excited by the nanophotonic device in use. This gives an enhancement factor in excitation of $|E|^2$, where E is the electric field amplitude felt by the molecule.

Moreover, in a second step, the radiation of the Raman scattering to the far-field also benefits from the nanophotonic device in use. This gives an additional approximated enhancement factor of $|E|^2$.

Approximating the excitation wavelength to the Raman scattering wavelength, this gives us the well known $|E|^4$ enhancement in SERS.

In their theoretical analysis, the authors only consider the enhancement in the excitation of the Chiral Molecule. Also considering the enhancement of the Raman Scattering in the presence of the metasurface would not be straight forward, because one would need to take into account the polarization state of the re-radiated fields (which does not necessarily have to maintain the incoming circular polarization). Nevertheless, in order to describe the enhancement of ROA, this consideration is not optional, but mandatory to get a complete theoretical description and adequate results.

- Checking the references of the article and performing some literature research I came to the conclusion that the bibliography in the article is far from being complete. The study of enhanced chiral light-matter interactions in dielectric structures was pioneered by the group of Jennifer Dionne at Stanford University, and I could not find a sole reference to their seminal articles in this field in the manuscript. Here I highlight some of the most relevant works, but there are more:

First proposals of dielectric structures as ideal platforms to enhance chiral light-matter interactions:

García-Etxarri, Aitzol, and Jennifer A. Dionne. "Surface-enhanced circular dichroism spectroscopy mediated by nonchiral nanoantennas." *Physical Review B* 87.23 (2013): 235409.

Ho, Chi-Sing, et al. "Enhancing enantioselective absorption using dielectric nanospheres." *ACS*

Photonics 4.2 (2017): 197-203.

The first proposal of dielectric metasurfaces made of disks as platforms to enhance chiral light-matter interactions:

Solomon, Michelle L., et al. "Enantiospecific optical enhancement of chiral sensing and separation with dielectric metasurfaces." ACS Photonics 6.1 (2018): 43-49.

An engineered metasurface design to get much higher chiral enhancement factors that the authors could also use in future developments in their technique

Hu, Jack, Mark Lawrence, and Jennifer A. Dionne. "High-quality factor dielectric metasurfaces for ultraviolet circular dichroism spectroscopy." ACS Photonics 7.1 (2019): 36-42.

A similar experimental development, using fluorescence detected CD instead of ROA

Solomon, Michelle L., et al. "Fluorescence-Detected Circular Dichroism of a Chiral Molecular Monolayer with Dielectric Metasurfaces." arXiv preprint arXiv:2008.11270 (2020).

· Lastly, there is one thing I don't understand in Figs. 3-4:

Why is the maximum value of both Raman and ROA ~ 1 ? I suspect that some normalization is being performed in these spectra but it is not mentioned in the main text. Also, how does this supposed normalization affect the enhanced spectra?

Did the authors consider including a ROA/Raman enhancement spectra dividing the results in the metasurface by those in silica?

In summary, I think the article needs major revisions before it is ready to be accepted. Nevertheless, I do think that the experimental relevance of the technique is very high. So, if all the aforementioned points would be addressed properly (especially the one on the theoretical interpretation), I would recommend its acceptance in Nature Communications.

Reviewer #3 (Remarks to the Author):

This is a very well constructed and described study on a challenging problem. The application of superchiral fields for enhanced sensitivity of detection of chiral molecules using absorbance measurements, i.e. electronic circular dichroism, has attracted considerable interest over the last decade or so. While those studies have demonstrated the potential of chiroptical detection, the limited information obtainable due to the small number of electronic transitions being measured is a limitation. Goda et al. have now changed that by successfully combining the enhancement provided by superchiral fields that can be generated by nanosculptured surfaces, with the unrivalled sensitivity to stereochemistry of ROA spectroscopy.

The authors have taken a different approach to enhancing ROA signals to the published studies that have explored more standard plasmon resonance processes. The authors have suitably presented their methodology in context to these past studies, confirming the novelty of their strategy of using anapoles from Si nanodisks. The SI file provides a suitable theoretical analysis substantiating the design of their nanodisk surfaces and the determination of their enhancement factors as being $\sim 10^2$.

Most importantly for the chiroptical spectroscopy community, the authors have presented their ROA spectra measured from their homebuilt ROA spectrometer, as well as their superchiral field-enhanced

ROA spectra. Artefact control in ROA is an important issue particularly when claims of enhancement are being assessed, and based upon the presented spectra I am perfectly satisfied that these superchiral field enhanced spectra are correct signals and not artefacts. While the samples used are still highly concentrated (5M) and the real test will come when studying low concentration spectra, the presented spectra are high quality and sufficient for supporting the authors' claims.

I think this is a very exciting report that will spur further development of chiroptical spectroscopies and smarter design of nanostructured surfaces for enhancing optical phenomena. I am very happy to support publication of this paper in Nature Communications.

To Reviewer #1:

We are grateful to the Reviewer for taking the time to review our manuscript and give us his/her valuable comments. We have taken all the comments into consideration and have made appropriate changes to the manuscript. Our point-by-point response appears below, in which we first echo the Reviewer's comments (shown in *italic*) and then respond to them. All the changes to the manuscript and Supplementary Information are highlighted in red.

Reviewer #1's comment #1:

This paper represents a significant advance in ROA measurement methodology. It is shown by the use of a suitably constructed silicon nanodisk array that the Raman and ROA of molecules located in the chirally-enhanced near-field region of the array are enhanced approximately by a factor of 100. The symmetry of the nanodisk array gives rise to an anapole toroidal field which theoretically suppresses linear polarization bias of the sample that in turn allows transfer of RCP and LCP light from the far field to the enhanced near field without the introduction of ROA artifacts (however see below) and a preservation of the ratio of ROA to Raman intensities, the so-called circular intensity difference (CID). The demonstration of near-field Raman/ROA enhancement of the nanodisk array is made by comparison to the Raman/ROA far-field intensities obtained using a silicon disk of the same dimensions by lacking the array. This achievement is significant because it addresses in a fundamental way a general disadvantage of ROA, namely the small ratio of ROA to Raman intensities and the need to use high concentrations, relative high incident laser powers and long measurement times to achieve ROA spectra with good signal quality.

The paper is carefully presented with demonstrations, including critically the SI material, of several key steps including 1) the symmetry properties of the array, 2) the anapole nature of the chiral enhancing near field, 3) the effect of this field on the induced electric dipole and induced magnetic dipole moments, including the optical chirality of the incident light that in this case, contributes to the generation of the observed enhanced Raman and ROA, 4) the Raman and ROA enhancement factors for the nanodisk array and 5) the dissymmetry factor achieved by the array showing the preservation of the chiral light environment from the far field to the near field and the preservation of the CID of the ROA.

Authors' response:

We thank the Reviewer for the positive comment.

Reviewer #1's comment #2:

Fig 1a shows incident CP laser radiation along the vertical z-axis with Raman scattering emerging apparently in any direction in the xy-plane just above the nanodisk array, i.e. 90-degree ICP-ROA scattering, yet the instrument layout in Fig. 2d shows both the incident and 180-degree backscattering ICP-ROA occurring along the vertical z-axis and only subsequently reflected into the horizontal direction by a mirror. This needs to be corrected or clarified.

Authors' response:

We thank the Reviewer for the suggestion, which we agree with. To address his/her comment, we have revised Fig. 1a as seen on the right. It now

Fig. R1. Revised Fig. 1a

shows a 180-degree backscattering ICP-ROA setup.

Reviewer #1's comment #3:

There are several issues with Fig. 3 containing chirally-enhanced near field and expanded far field Raman and ROA of both enantiomers of alpha-pinene that need comments by the authors. The Raman and ROA of neat alpha-pinene are extremely well-known in the literature and used extensively to calibrate standard ROA spectrometers. The accuracy and artifact level of the ROA can be assessed for example by comparison to the paper by W. Hug, Appl. Spectrosc. 57, 1-13 (2003) where the concept of virtual enantiomer is presented for the first time, and upon which the instrument design is based but only a secondary reference 29 is provided. In Figs. 9 and 10 of the Hug paper high-quality conventional Raman and ROA spectra of both enantiomers of alpha pinene are displayed. Compared to the corresponding Raman and ROA spectra in Fig. 3 of the present paper allows one to verify all the features of the Raman spectra and roughly verify the signs and magnitudes of the ROA bands. The problem with the ROA comparison is the relative low signal to noise ratio and the crowding, almost overlapping, of the ROA bands compared to the corresponding Raman bands. Such crowding does not occur in conventional ROA spectra as in, for example, the Hug paper. The Hug spectra have signal to noise ratios approaching 100 and much higher apparent spectral resolution, 33 minutes of collection time and 420 mW incident laser power using a 35 microliter cell. The ROA spectra in Fig. 3 here have signal to noise ratio closer to only 10, 300 minutes of collection time (5h), 800 mW of laser power, and an effective volume of 0.1 picoliters. The only favorable aspect of this comparison is the effective volume of 0.1 pL that arises from the 100 fold intensity enhancement, but beyond that there is no demonstrated advantage of the superchiral ROA enhancement. The authors should be more forthcoming about the quality of the ROA spectra they present versus the quality available from existing conventional ROA methodology.

Authors' response:

We thank the Reviewer for the comments. First, we agree that the reference (W. Hug, Appl. Spectrosc. 57, 1-13 (2003)) mentioned by the Reviewer presented the concept of virtual enantiomer for the first time. To address the Reviewer's comment, we have added this reference to our revised manuscript. Second, we agree that the signal-to-noise ratio in our measurement is relatively low, but this mainly results from the small effective volume (0.1 pL) of our measured sample and the moderate signal-collection efficiency of our homemade ROA setup that limit the measured ROA signal intensity in a specific integration time. The crowding mentioned by the Reviewer is also due to the relatively low signal-to-noise ratio. To compare our method with the conventional methodology, we think it is fairer to compare the quality of ROA spectra at the same experimental conditions. In fact, the quality of ROA spectra measured by our method is better than that of the conventional ROA methodology at the same experimental conditions when the measured sample volume or thickness is very small. This is verified by our experiment result shown in Supplementary Fig. 2. The quality of measured ROA spectrum on the silicon nanodisk array (Supplementary Fig. 2c) which represents our method is better than that on the silica substrate (Supplementary Fig. 2d) which represents the conventional ROA methodology. A further improved signal-to-noise ratio can be expected by our method if an optimized commercial ROA instrument is used.

Reviewer #1's comment #4:

A second issue the authors need to address is their statement on line 140 of the "excellent artifact-free condition of the ROA spectra in Fig. 3. This is not correct. The most difficult problem in ROA measurement is to avoid artifacts in highly polarized Raman bands. Alpha pinene has two such bands, one at 671 cm⁻¹ and the

other at 1663 cm^{-1} . The latter is not shown in the Raman and ROA spectra in Figs. 3a and 3b, respectively; however the very strong Raman band at 671 cm^{-1} is accompanied by a very strong, presumably artifact, ROA band, the strongest in the ROA spectrum by a factor of ~ 2 . According to the ROA literature in general and the Hug paper in particular instead of a strong ROA band 671 cm^{-1} there should be one of the smallest ROA bands, smaller than its neighboring band near 600 cm^{-1} . Most likely there would also be a strong ROA artifact band at 1663 cm^{-1} if this were shown. The authors need to confront this artifact problem which calls into question the artifact-free contention of these new ROA spectra.

Authors' response:

We thank the Reviewer for the comment, which we agree with. Regarding the artifact, we think it mainly came from our home-made ROA system rather than our silicon nanodisk array as we also observed a very similar artifact of this ROA band measured on the silica substrate. To address the Reviewer's comment and validate our claim, we have revised it from "excellent artifact-free condition" to "artifact-suppressed condition" in the revised manuscript (page 4, paragraphs 2-3).

Reviewer #1's comment #5:

Third, the authors need to be more clear about the comparison of superchiral enhanced spectra displayed in Fig. 3, versus the reference non-enhanced spectra shown as zoomed enlarged spectra by a factor of approximately 100. As shown in the SI, the superchiral enhanced spectra are not direct measurements but rather spectral subtractions to remove the larger contribution from the superimposed far field spectra. This fact explains why the superchiral enhanced ROA spectra have noise levels similar to the non-enhanced spectra. The authors need to make these facts more clear and address the question of whether the superchiral near-field Raman/ROA can be seen directly without interference from the far field contributions.

Authors' response:

We thank the Reviewer for the comment, which we agree with. To address it, we have added the following text to the revised manuscript (page 4, paragraph 3): "a Raman signal enhancement with an average enhancement factor of $\sim 10^2$ in the near-field region, which was obtained by removing the far-field contribution, was evident on the silicon nanodisk array (see "Raman and ROA enhancement factors achieved by the silicon nanodisk array" in Supplementary Information for details)." Also, we have added the following text to the revised manuscript (page 4, paragraph 3): "Similar to the Raman spectra, an ROA signal enhancement with an average enhancement factor of $\sim 10^2$ in the near-field region, which was obtained by removing the far-field contribution, was evident for both (\pm)-alpha-pinene (see "Raman and ROA enhancement factors achieved by the silicon nanodisk array" in Supplementary Information for details).

Next, the superchiral near-field Raman/ROA is difficult to observe without interference from the far-field contribution. This is mainly due to the technical difficulty in preparing an ultrathin alpha-pinene (volatile liquid) film with the thickness of the near-field region ($\sim 250\text{ nm}$) that is perfectly sealed on our chip and does not evaporate in the long-time measurement (several hours). Although our measurement is not direct, we think it is sufficient to verify the observation and extract the enhancement factors of the superchiral near-field Raman/ROA measurements. To address the Reviewer's comment, we have added the following text to the revised Supplementary Information (page 8, paragraph 2): "It is important to note that it is technically difficult to prepare an ultrathin alpha-pinene (volatile liquid) film with the thickness of the near-field region ($\sim 250\text{ nm}$) that is perfectly sealed on our chip and does not evaporate in the long-time measurement (several hours) for directly measuring the near-field Raman and ROA signals."

Reviewer #1's comment #6:

In summary, the authors have demonstrated the ability to isolate and measure superchiral enhanced Raman and ROA spectra that are 100-times enhanced by the use of nanodisk array. This is a conceptual and experimentally-realized accomplishment of significant interest. There is, nevertheless, still an unresolved problem of a very large artifact ROA band at 671 cm^{-1} that should be at least an order of magnitude smaller. Although the superchiral enhanced ROA emerges from a tiny volume of less than 1 pL, because of the presence of larger far-field Raman and ROA spectral contributions, there is still no noise improvement even though there is a 100-fold enhancement due to the superchiral fields provided by the nanodisk array. Thus, there are several issues that need to be resolved or made more clear before the paper can be accepted for publication.

Authors' response:

We thank the Reviewer for the comment and encouragement. We humbly hope that he/she is happy with our point-by-point response to his/her comments.

To Reviewer #2:

We are grateful to the Reviewer for taking the time to review our manuscript and give us his/her valuable comments. We have taken all the comments into consideration and have made appropriate changes to the manuscript. Our point-by-point response appears below, in which we first echo the Reviewer's comments (shown in italic) and then respond to them. All the changes to the manuscript and Supplementary Information are highlighted in red.

Reviewer #2's comment #1:

I read the manuscript by Ting-Hui Xiao et.al. with great care. It presents a very promising study on the enhancement of the ROA of chiral molecules mediated by all-dielectric metasurfaces. The experimental results are very promising, and the technological implications of the technique are of high relevance. Nevertheless, I encountered several drawbacks in the analysis of the results of the article. I think that the article could deserve its publication in Nature Communications, but not in this form. I would recommend its acceptance if the authors would address the following points convincingly

Authors' response:

We thank the Reviewer for the positive comment.

Reviewer #2's comment #2:

An anapole ideally does not radiate to the far-field. Thus, one cannot couple to it with far-field excitations either. Although the authors mention that the cancelation of toroidal and magnetic dipolar modes in their system is not ideal, they should develop and clarify this point in the main text. Also, this far from perfect cancelation, makes me think if this "mode" deserves to be called an anapolar mode. This criticism points more to a linguistic argument, it does not preclude the experimental results from being valid. Nevertheless, I would suggest the authors to rephrase a bit the words used in their theoretical interpretation. I would not refer to these resonances as anapolar modes.

Authors' response:

We thank the Reviewer for the comment. We agree that the use of the term "anapole mode" is not completely accurate as the cancelation of the far-field radiation is not ideal. We think it is more accurate to call it a hybrid mode as the excited mode is a hybridization of electric and toroidal dipole modes. To address the Reviewer's comment, we have replaced the "anapole mode" with the "hybrid mode" throughout the manuscript and Supplementary Information.

Reviewer #2's comment #3:

As the authors mention in the manuscript, Silicon is a lossy material In the frequency of their excitation (532nm). In the introduction, they say that heating can perturb the ROA signal of chiral molecules. Although apparently this is not a serious issue in their article (the features of the ROA spectra in their metasurface are similar to that obtained in silica), I miss an analysis of the heating in their system in comparison with other previously proposed plasmonic systems to quantify the adequateness of their experimental platform.

Authors' response:

We thank the Reviewer for the comment. We agree that a quantitative analysis of photothermal heating in our

silicon nanodisk array in comparison with its plasmonic counterpart is helpful. To address the comment, we have used the finite element method (FEM) to simulate temperature distributions of both the silicon nanodisk array and its plasmonic counterpart with an identical incident power density of $5 \text{ mW}/\mu\text{m}^2$ in a thermal equilibrium state. The wavelength of incident light is 532 nm while the environmental temperature is 25 °C. The plasmonic counterpart has an identical geometric structure to that of the silicon nanodisk array, but is made of silver. It is evident from the simulation result shown in Fig. R2 that the silicon nanodisk array has a lower temperature than its plasmonic counterpart under the identical experimental conditions, which verifies the relatively low photothermal heat generation in the silicon nanodisk array. We have added Fig. R2 and the following text to the revised Supplementary Information (page 6, paragraph 2): “To show low photothermal heating in the silicon nanodisk array in comparison with its plasmonic counterpart, we used the finite element method (FEM) to simulate temperature distributions of both the silicon nanodisk array and its plasmonic counterpart with an identical incident power density of $5 \text{ mW}/\mu\text{m}^2$ in a thermal equilibrium state. The wavelength of the incident light is 532 nm while the environmental temperature is 25 °C. The plasmonic counterpart has an identical geometric structure to that of the silicon nanodisk array, but is made of silver. It is evident from the simulation result shown in Supplementary Fig. 2 that the silicon nanodisk array has a lower temperature than its plasmonic counterpart under the identical experimental conditions, which verifies the relatively low photothermal heat generation in the silicon nanodisk array.” Also, to validate our claim, we have also revised the relevant text in the manuscript (page 5, paragraph 2): “low photothermal heat generation”.

Fig. R2 Temperature distributions of the silicon nanodisk array and its plasmonic counterpart with an identical incident power density of $5 \text{ mW}/\mu\text{m}^2$ in a thermal equilibrium state.

Reviewer #2’s comment #4:

I have serious doubts about the theoretical interpretation of the ROA enhancement they develop in the supplementary information, which I consider very simplistic and incomplete.

In simple words, Surface Enhanced Raman Spectroscopy (SERS), the increase in the Raman signal

encompasses two different sources of enhancement. On the one hand, the molecules under study are more efficiently excited by the nanophotonic device in use. This gives an enhancement factor in excitation of $|E|^2$, where E is the electric field amplitude felt by the molecule.

Moreover, in a second step, the radiation of the Raman scattering to the far-field also benefits from the nanophotonic device in use. This gives an additional approximated enhancement factor of $|E|^2$.

Approximating the excitation wavelength to the Raman scattering wavelength, this gives us the well known $|E|^4$ enhancement in SERS.

In their theoretical analysis, the authors only consider the enhancement in the excitation of the Chiral Molecule. Also considering the enhancement of the Raman Scattering in the presence of the metasurface would not be straight forward, because one would need to take into account the polarization state of the re-radiated fields (which does not necessarily have to maintain the incoming circular polarization). Nevertheless, in order to describe the enhancement of ROA, this consideration is not optional, but mandatory to get a complete theoretical description and adequate results.

Authors' response:

We thank the Reviewer for the comment. We agree that the theoretical interpretation of the ROA enhancement is somewhat simplistic. We also agree that SERS includes two steps. The first step is to enhance the localized electric field to increase the Raman excitation of molecules while the second step is to enhance the radiation of Raman scattering. To address the Reviewer's comment and consider both steps for making our theoretical framework more complete, we have added the following text and new Supplementary Fig. 1 to the Supplementary Information (page 3, paragraph 1):

“As Raman scattering consists of two processes, namely, the excitation process and the radiation process, the electric and magnetic dipole moments that correspond to the above two processes can be described as follows. For the excitation process, the electric and magnetic dipole moments can be written as

$$\widehat{\boldsymbol{\mu}}^e = \widehat{\alpha}^e \widehat{\mathbf{E}}^e + \frac{1}{3} \widehat{A}_Y^e \nabla \widehat{E}_Y^e - i \widehat{G}^e \widehat{\mathbf{B}}^e, \quad (10)$$

$$\widehat{\mathbf{m}}^e = \widehat{\chi}^e \widehat{\mathbf{B}}^e + i \widehat{G}^e \widehat{\mathbf{E}}^e, \quad (11)$$

where the superscript e indicates the physical variables of the excitation process. For the excitation rate R^e of Raman scattering, we have

$$R^e \propto \langle \mathbf{E}^e \cdot \boldsymbol{\mu}^e + \mathbf{B}^e \cdot \dot{\mathbf{m}}^e \rangle = \frac{\omega^e}{2} \left(\widehat{\mathbf{E}}^{e*} \cdot \widehat{\boldsymbol{\mu}}^e + \widehat{\mathbf{B}}^{e*} \cdot \widehat{\mathbf{m}}^e \right), \quad (12)$$

where the brackets indicate average values over time, \mathbf{E}^e , \mathbf{B}^e , $\boldsymbol{\mu}^e$, and \mathbf{m}^e are the time-dependent real parts of $\widehat{\mathbf{E}}^e$, $\widehat{\mathbf{B}}^e$, $\widehat{\boldsymbol{\mu}}^e$, and $\widehat{\mathbf{m}}^e$, respectively, ω^e is the angular frequency, and the hat indicates the time-dependent imaginary part of each parameter. By substituting Eqs. (10) and (11) into Eq. (12), the excitation rate can be written as

$$R^e \propto \frac{\omega^e}{2} \left[\widehat{\alpha}^e |\widehat{\mathbf{E}}^e|^2 + \frac{1}{3} \widehat{E}_Y^{e*} \cdot \widehat{A}_Y^e \nabla \widehat{E}_Y^e + \widehat{\chi}^e |\widehat{\mathbf{B}}^e|^2 \right] + \omega^e \widehat{G}^e \left(\widehat{\mathbf{E}}^{e*} \cdot \widehat{\mathbf{B}}^e \right), \quad (13)$$

where $\widehat{\alpha}^e$, \widehat{A}_Y^e , $\widehat{\chi}^e$ and \widehat{G}^e are the time-dependent imaginary parts of $\widehat{\alpha}^e$, \widehat{A}_Y^e , $\widehat{\chi}^e$ and \widehat{G}^e . Based on the definition of optical chirality, the optical chirality of the electromagnetic field in the excitation process C^e is given by

$$C^e = \frac{\varepsilon_0}{2} \mathbf{E}^e \cdot \nabla \times \mathbf{E}^e + \frac{1}{2\mu_0} \mathbf{B}^e \cdot \nabla \times \mathbf{B}^e. \quad (14)$$

Using the Maxwell equations, $\nabla \times \mathbf{E}^e = i\omega^e \mathbf{B}^e$ and $\nabla \times \mathbf{B}^e = -i\omega^e \mu_0 \varepsilon_0 \mathbf{E}^e$, the optical chirality can be rewritten as

$$C^e = \frac{\varepsilon_0}{2} (\mathbf{B}^e \cdot \mathbf{E}^e - \mathbf{E}^e \cdot \mathbf{B}^e) = \frac{\varepsilon_0 \omega}{2} (\widehat{\mathbf{E}}^{e*} \cdot \widehat{\mathbf{B}}^e). \quad (15)$$

By substituting Eq. (15) into Eq. (13), the excitation rate is found to be

$$R^e \propto \frac{\omega^e}{2} \left[\widehat{\alpha}^e |\widehat{\mathbf{E}}^e|^2 + \frac{1}{3} \widehat{\mathbf{E}}^{e*} \cdot \widehat{A}_Y^e \nabla \widehat{E}_Y^e + \widehat{\chi}^e |\widehat{\mathbf{B}}^e|^2 \right] + \frac{2C^e \widehat{G}^e}{\varepsilon_0}. \quad (16)$$

If we express the middle term in the brackets as $\frac{1}{3} \widehat{\mathbf{E}}^{e*} \cdot \widehat{A}_Y^e \nabla \widehat{E}_Y^e = T_s^e + T_{as}^e$, where T_s^e is the chirally symmetric part while T_{as}^e is the chirally antisymmetric part, Eq. (16) can be expressed as

$$R^e \propto \frac{\omega^e}{2} \left(\widehat{\alpha}^e |\widehat{\mathbf{E}}^e|^2 + T_s^e + \widehat{\chi}^e |\widehat{\mathbf{B}}^e|^2 \right) + \frac{\omega^e T_{as}^e}{2} + \frac{2C^e \widehat{G}^e}{\varepsilon_0}, \quad (17)$$

where the first term on the right side is chirally symmetric while the last two terms are chirally antisymmetric. Thus, for a chiral molecule, its excitation rates with opposite optical chirality values can be expressed as

$$R_+^e \propto \frac{\omega^e}{2} \left(\widehat{\alpha}^e |\widehat{\mathbf{E}}^e|^2 + T_s^e + \widehat{\chi}^e |\widehat{\mathbf{B}}^e|^2 \right) + \frac{\omega^e T_{as}^e}{2} + \frac{2C^e \widehat{G}^e}{\varepsilon_0}, \quad (18)$$

$$R_-^e \propto \frac{\omega^e}{2} \left(\widehat{\alpha}^e |\widehat{\mathbf{E}}^e|^2 + T_s^e + \widehat{\chi}^e |\widehat{\mathbf{B}}^e|^2 \right) - \frac{\omega^e T_{as}^e}{2} - \frac{2C^e \widehat{G}^e}{\varepsilon_0}. \quad (19)$$

For the radiation process, the electric and magnetic dipole moments can be written as

$$\widehat{\boldsymbol{\mu}}^r = \widehat{\alpha}^r \widehat{\mathbf{E}}^r + \frac{1}{3} \widehat{A}_Y^r \nabla \widehat{E}_Y^r - i \widehat{G}^r \widehat{\mathbf{B}}^r, \quad (20)$$

$$\widehat{\mathbf{m}}^r = \widehat{\chi}^r \widehat{\mathbf{B}}^r + i \widehat{G}^r \widehat{\mathbf{E}}^r, \quad (21)$$

where the superscript r indicates the physical variables of the radiation process. For the radiation rate R^e of Raman scattering, we have

$$R^r \propto \langle \mathbf{E}^r \cdot \widehat{\boldsymbol{\mu}}^r + \mathbf{B}^r \cdot \widehat{\mathbf{m}}^r \rangle = \frac{\omega^r}{2} (\widehat{\mathbf{E}}^{r*} \cdot \widehat{\boldsymbol{\mu}}^r + \widehat{\mathbf{B}}^{r*} \cdot \widehat{\mathbf{m}}^r). \quad (22)$$

As Eq. (22) has an identical mathematical form to Eq. (12), we can obtain the radiation rate by following the similar deduction from Eq. (13) to Eq. (16). Then, the radiation rate can be expressed as

$$R^r \propto \frac{\omega^r}{2} \left(\widehat{\alpha}^r |\widehat{\mathbf{E}}^r|^2 + \frac{1}{3} \widehat{\mathbf{E}}^{r*} \cdot \widehat{A}_Y^r \nabla \widehat{E}_Y^r + \widehat{\chi}^r |\widehat{\mathbf{B}}^r|^2 \right) + \frac{2C^r \widehat{G}^r}{\varepsilon_0}. \quad (23)$$

As the radiation process is related with the excitation process, we define C_+^r as the optical chirality of radiation $\frac{1}{3} \widehat{\mathbf{E}}_+^{r*} \cdot \widehat{A}_{Y+}^r \nabla \widehat{E}_{Y+}^r = T_+$ for the excitation rate of R_+^e as well as C_-^r as the optical chirality of radiation $\frac{1}{3} \widehat{\mathbf{E}}_-^{r*} \cdot \widehat{A}_{Y-}^r \nabla \widehat{E}_{Y-}^r = T_-$ for the excitation rate of R_-^e . Thus, the radiation rates for the excitation rates of R_+^e and R_-^e can be written respectively as

$$R_+^r \propto \frac{\omega^r}{2} \left(\widehat{\alpha}^r |\widehat{\mathbf{E}}^r|^2 + T_+ + \widehat{\chi}^r |\widehat{\mathbf{B}}^r|^2 \right) + \frac{2C_+^r \widehat{G}^r}{\varepsilon_0}, \quad (24)$$

$$R_-^r \propto \frac{\omega^r}{2} \left(\widehat{\alpha}^r |\widehat{\mathbf{E}}^r|^2 + T_- + \widehat{\chi}^r |\widehat{\mathbf{B}}^r|^2 \right) + \frac{2C_-^r \widehat{G}^r}{\varepsilon_0}. \quad (25)$$

Then, the ROA intensity is found to be

$$R_+^e R_+^r - R_-^e R_-^r \propto \left[\frac{\omega^e}{2} \left(\widehat{\alpha}^e |\widehat{\mathbf{E}}^e|^2 + T_s^e + \widehat{\chi}^e |\widehat{\mathbf{B}}^e|^2 \right) + \frac{\omega^e T_{as}^e}{2} + \frac{2C^e \widehat{G}^e}{\varepsilon_0} \right] \left[\frac{\omega^r}{2} \left(\widehat{\alpha}^r |\widehat{\mathbf{E}}^r|^2 + T_+ + \widehat{\chi}^r |\widehat{\mathbf{B}}^r|^2 \right) + \frac{2C_+^r \widehat{G}^r}{\varepsilon_0} \right] - \left[\frac{\omega^e}{2} \left(\widehat{\alpha}^e |\widehat{\mathbf{E}}^e|^2 + T_s^e + \widehat{\chi}^e |\widehat{\mathbf{B}}^e|^2 \right) - \frac{\omega^e T_{as}^e}{2} - \frac{2C^e \widehat{G}^e}{\varepsilon_0} \right] \left[\frac{\omega^r}{2} \left(\widehat{\alpha}^r |\widehat{\mathbf{E}}^r|^2 + T_- + \widehat{\chi}^r |\widehat{\mathbf{B}}^r|^2 \right) + \frac{2C_-^r \widehat{G}^r}{\varepsilon_0} \right]. \quad (26)$$

To understand the underlying physics of superchiral-field-enhanced ROA, we need to consider Eq. (26) under some approximations for simplification. For most chiral molecules in a superchiral field with a strong enhancement of optical chirality, especially for those composed of idealized axially symmetric bonds⁶, the signal intensity contribution from the electric-field gradient ωT is much smaller than that from the optical chirality $2C\widehat{G}/\varepsilon_0$. Moreover, for non-magnetic chiral molecules, $\widehat{\chi} |\widehat{\mathbf{B}}|^2$ is much smaller than $\widehat{\alpha} |\widehat{\mathbf{E}}|^2$. Thus, Eq. (26) can be simplified to

$$R_+^e R_+^r - R_-^e R_-^r \propto \left[\frac{\omega^e \widehat{\alpha}^e |\widehat{\mathbf{E}}^e|^2}{2} + \frac{2C^e \widehat{G}^e}{\varepsilon_0} \right] \left[\frac{\omega^r \widehat{\alpha}^r |\widehat{\mathbf{E}}^r|^2}{2} + \frac{2C_+^r \widehat{G}^r}{\varepsilon_0} \right] - \left[\frac{\omega^e \widehat{\alpha}^e |\widehat{\mathbf{E}}^e|^2}{2} - \frac{2C^e \widehat{G}^e}{\varepsilon_0} \right] \left[\frac{\omega^r \widehat{\alpha}^r |\widehat{\mathbf{E}}^r|^2}{2} + \frac{2C_-^r \widehat{G}^r}{\varepsilon_0} \right]. \quad (27)$$

Additionally, for the superchiral-field enhanced ROA based on our silicon nanodisk array, the optical chirality

C^e at the wavelength of excitation light is precisely controlled and enhanced while the optical chirality C^r of radiation at the wavelength of Raman scattering is significantly decayed and is approximately equal to zero in the near-field region (Supplementary Fig. 1). The optical reciprocity theorem is used here to calculate the optical chirality C^r . Specifically, to estimate the far-field radiation enhancement induced by the near-field enhancement, a far-field excitation to calculate the enhanced near-field is used. The circularly polarized light is used as the far-field excitation to calculate the enhanced near-field as light with any possible polarization can be written as a linearly combination of left and right circularly polarized light while it follows the rotational symmetry of the excitation process. It is noted that we do not need to consider the polarization of the enhanced far-field radiation or the re-radiated field from the silicon nanodisk array because the ROA in our experiments is the incident circularly polarized ROA (ICP-ROA), in which only the polarization of incident light needs to be considered in the excitation process while the polarization of Raman scattering does not need to be considered in the detection process. With the above approximation, Eq. (27) can be further simplified to

$$R_+^e R_+^r - R_-^e R_-^r \propto \left[\frac{\omega^e \widehat{\alpha}^e |\widehat{E}^e|^2}{2} + \frac{2C^e \widehat{G}^e}{\varepsilon_0} \right] \left[\frac{\omega^r \widehat{\alpha}^r |\widehat{E}^r|^2}{2} \right] - \left[\frac{\omega^e \widehat{\alpha}^e |\widehat{E}^e|^2}{2} - \frac{2C^e \widehat{G}^e}{\varepsilon_0} \right] \left[\frac{\omega^r \widehat{\alpha}^r |\widehat{E}^r|^2}{2} \right] = \frac{2\omega^r \widehat{\alpha}^r \widehat{G}^e C^e |\widehat{E}^r|^2}{\varepsilon_0}. \quad (28)$$

Based on Eq. (28), the circular intensity difference (CID) of the ROA can be expressed as

$$\text{CID} = \frac{I_{RCP} - I_{LCP}}{I_{RCP} + I_{LCP}} \propto \frac{R_+^e R_+^r - R_-^e R_-^r}{R_+^e R_+^r + R_-^e R_-^r} \propto \frac{8\widehat{G}^e C^e}{\varepsilon_0 \omega^e \widehat{\alpha}^e |\widehat{E}^e|^2}, \quad (29)$$

where I_{RCP} and I_{LCP} are the intensities of Raman scattering excited by incident RCP and LCP light, respectively.”

Supplementary Fig. 1 | **a**, Average optical activity (RCP excitation) in the near-field region on the silicon nanodisk array as a function of wavelength, normalized by the optical chirality of incident CPL in the far field. **b**, Averaged square of the electric-field magnitude in the near-field region on the silicon nanodisk array as a function of wavelength, normalized by the square of the electric-field magnitude of incident CPL in the far field.

Furthermore, we have revised the relevant text in Supplementary Information as follows.

Supplementary Information (page 9, paragraph 1):

“In order to compare the experimental results with the theoretical results, we calculated the theoretical Raman and ROA enhancement factors in the near-field region. Based on Eq. (29), the theoretical Raman enhancement factor at a specific point in the near-field region is found to be

$$EF_{Raman} = \frac{I_{RCP} + I_{LCP}}{I_{ORCP} + I_{OLCP}} = \frac{R_+^e R_+^r + R_-^e R_-^r}{R_{0+}^e R_{0+}^r + R_{0-}^e R_{0-}^r} = \frac{|\widehat{E}^e|^2}{|\widehat{E}_0^e|^2} \left| \frac{\widehat{E}^r}{\widehat{E}_0^r} \right|^2, \quad (31)$$

where the subscript 0 indicates the physical variables without the superchiral-field enhancement. The theoretical Raman enhancement factor in the near-field region can be analytically calculated to be

$$EF_{Raman} = \frac{\int_{V_d}^{V_n} \frac{|\widehat{E}^e|^2}{|\widehat{E}_0^e|^2} \left| \frac{\widehat{E}^r}{\widehat{E}_0^r} \right|^2 dv}{\int_{V_d}^{V_n} dv}. \quad (32)$$

It is noted that $\left| \frac{\widehat{E}^r}{\widehat{E}_0^r} \right|^2$ which corresponds to the near-field enhancement is approximately equal to the far-field radiation enhancement at the excitation wavelength of Raman scattering due to the optical reciprocity theorem. Based on this equation, the theoretical Raman enhancement factor in the near-field region is estimated to be 93, which is on the order of $\sim 10^2$ and agrees with our experimental result. Similarly based on Eq. (28), the theoretical ROA enhancement factor at a specific point in the near-field region is found to be

$$EF_{ROA} = \frac{I_{RCP} - I_{LCP}}{I_{ORCP} - I_{OLCP}} = \frac{R_+^e R_+^r - R_-^e R_-^r}{R_{0+}^e R_{0+}^r - R_{0-}^e R_{0-}^r} = \frac{C^e}{C_0^e} \left| \frac{\widehat{E}^r}{\widehat{E}_0^r} \right|^2. \quad (33)$$

It is evident from Eq. (33) that the silicon nanodisk array works as a superchiral-field generator to enhance the chiral light-molecule interaction to enhance the Raman intensity difference between LCP and RCP excitation in the excitation process while it works as an optical antenna to enhance the far-field radiation to further enhance the Raman intensity difference between LCP and RCP excitation in the radiation process. The theoretical ROA enhancement factor in the near-field region can be analytically calculated by

$$EF_{ROA} = \frac{\int_{V_d}^{V_n} \frac{C^e}{C_0^e} \left| \frac{\widehat{E}^r}{\widehat{E}_0^r} \right|^2 dv}{\int_{V_d}^{V_n} dv}. \quad (34)$$

Based on this equation, the theoretical ROA enhancement factor in the near-field region is estimated to be 102, which is also on the order of $\sim 10^2$ and agrees with our experimental result. To further verify the theoretical ROA enhancement factor, we also evaluate the theoretical dissymmetric factor in the silicon nanodisk array, which determines the enhancement factor of the CID. As the estimated dissymmetric factor in the silicon nanodisk array is ~ 1 (See Supplementary Section 9 for details), the CID enhancement factor is ~ 1 , which means the theoretical ROA enhancement factor is almost identical to the theoretical Raman enhancement factor, which also agrees well with the above result.”

Supplementary Information (page 10, paragraph 2):

“The dissymmetric factor indicates the strength of chiral light-matter interaction based on intensity-normalized light, which excludes the influence of light intensity by normalization. This parameter predominantly determines the CID of ROA measurements. Based on Eq. (29), we have

$$CID = \frac{I_{RCP} - I_{LCP}}{I_{RCP} + I_{LCP}} \propto \frac{R_+^e R_+^r - R_-^e R_-^r}{R_+^e R_+^r + R_-^e R_-^r} \propto \frac{8\widehat{G}^e C^e}{\varepsilon_0 \omega^e \widehat{\alpha}^e |\widehat{E}^e|^2} \propto g. \quad (35)$$

As \widehat{G}^e , ε_0 , $\widehat{\alpha}^e$, and ω^e are constants for a molecule excited by light with a fixed frequency, the normalized circular excitation rate difference $\frac{R_+ - R_-}{R_+ + R_-}$ is determined by $C^e / |\widehat{E}^e|^2$ and ends up as the dissymmetric factor g .”

Reviewer #2's comment #5:

Checking the references of the article and performing some literature research I came to the conclusion that the bibliography in the article is far from being complete. The study of enhanced chiral light-matter interactions in dielectric structures was pioneered by the group of Jennifer Dionne at Stanford University, and

I could not find a sole reference to their seminal articles in this field in the manuscript. Here I highlight some of the most relevant works, but there are more:

First proposals of dielectric structures as ideal platforms to enhance chiral light-matter interactions:

*García-Etxarri, Aitzol, and Jennifer A. Dionne. "Surface-enhanced circular dichroism spectroscopy mediated by nonchiral nanoantennas." *Physical Review B* 87.23 (2013): 235409.*

*Ho, Chi-Sing, et al. "Enhancing enantioselective absorption using dielectric nanospheres." *ACS Photonics* 4.2 (2017): 197-203.*

The first proposal of dielectric metasurfaces made of disks as platforms to enhance chiral light-matter interactions:

*Solomon, Michelle L., et al. "Enantiospecific optical enhancement of chiral sensing and separation with dielectric metasurfaces." *ACS Photonics* 6.1 (2018): 43-49.*

An engineered metasurface design to get much higher chiral enhancement factors that the authors could also use in future developments in their technique

*Hu, Jack, Mark Lawrence, and Jennifer A. Dionne. "High-quality factor dielectric metasurfaces for ultraviolet circular dichroism spectroscopy." *ACS Photonics* 7.1 (2019): 36-42.*

A similar experimental development, using fluorescence detected CD instead of ROA

*Solomon, Michelle L., et al. "Fluorescence-Detected Circular Dichroism of a Chiral Molecular Monolayer with Dielectric Metasurfaces." *arXiv preprint arXiv:2008.11270* (2020).*

Authors' response:

We thank the Reviewer for the suggestion. To address it, we have added all the references suggested by the Reviewer to our revised manuscript.

Reviewer #2's comment #6:

Lastly, there is one thing I don't understand in Figs. 3-4: Why is the maximum value of both Raman and ROA ~ 1 ? I suspect that some normalization is being performed in these spectra but it is not mentioned in the main text. Also, how does this supposed normalization affect the enhanced spectra? Did the authors consider including a ROA/Raman enhancement spectra dividing the results in the metasurface by those in silica?

Authors' response:

We thank the Reviewer for the comment. As he/she correctly guessed, we normalized the maximum values of both Raman and ROA spectra measured on the silica substrate to ~ 1 . By using the above different normalizations for the Raman and ROA spectra, the values of Raman and ROA spectra measured on the silicon nanodisk array are also normalized. The normalizations do not affect the enhanced spectra as they are linear and identical on different substrates for both the Raman and ROA spectra. We did not include the ROA/Raman enhancement spectra that could be obtained by dividing the results in the silicon nanodisk array by those in

silica because the intensities of Raman and ROA spectra are zero or near-zero at some Raman shift values, making the normalized values highly sensitive to noise. To clarify this point, we have added the following text to the revised manuscript (page 4, paragraph 3): “Here the Raman spectra were normalized by the maximum Raman signal intensity value obtained on the silica substrate”; “Correspondingly, the measured ROA spectra of this pair of enantiomers on the silica substrate and silicon nanodisk array are shown in Fig. 3b in which the ROA spectra were normalized by the maximum ROA signal intensity value obtained on the silica substrate”; (page 5, paragraph 2): “Compared with the Raman spectra measured on the silica substrate, enhanced Raman spectra of both (±)-tartaric acid were measured on the silicon nanodisk array as shown in Fig. 4a in which the Raman spectra were normalized by the maximum Raman signal intensity value obtained on the silica substrate”; “More importantly, enhanced ROA signals were also observed on the silicon nanodisk array with an average enhancement factor of $\sim 10^2$ in the near-field region as shown in Fig. 4b in which the ROA spectra were normalized by the maximum ROA signal intensity value obtained on the silica substrate”.

Reviewer #2’s comment #7:

In summary, I think the article needs major revisions before it is ready to be accepted. Nevertheless, I do think that the experimental relevance of the technique is very high. So, if all the aforementioned points would be addressed properly (especially the one on the theoretical interpretation), I would recommend its acceptance in Nature Communications.

Authors’ response:

We thank the Reviewer for the positive comment. We have carefully addressed all the comments mentioned by the Reviewer in the above responses.

To Reviewer #3:

We are grateful to the Reviewer for taking the time to review our manuscript and give us his/her valuable comments. All the changes to the manuscript and Supplementary Information in relation to our response to the other two reviewers are highlighted in red.

Reviewer #3's comment #1:

This is a very well constructed and described study on a challenging problem. The application of superchiral fields for enhanced sensitivity of detection of chiral molecules using absorbance measurements, i.e. electronic circular dichroism, has attracted considerable interest over the last decade or so. While those studies have demonstrated the potential of chiroptical detection, the limited information obtainable due to the small number of electronic transitions being measured is a limitation. Goda et al. have now changed that by successfully combining the enhancement provided by superchiral fields that can be generated by nanosculptured surfaces, with the unrivalled sensitivity to stereochemistry of ROA spectroscopy.

The authors have taken a different approach to enhancing ROA signals to the published studies that have explored more standard plasmon resonance processes. The authors have suitably presented their methodology in context to these past studies, confirming the novelty of their strategy of using anapoles from Si nanodisks. The SI file provides a suitable theoretical analysis substantiating the design of their nanodisk surfaces and the determination of their enhancement factors as being $\sim 10^2$.

Most importantly for the chiroptical spectroscopy community, the authors have presented their ROA spectra measured from their homebuilt ROA spectrometer, as well as their superchiral field-enhanced ROA spectra. Artefact control in ROA is an important issue particularly when claims of enhancement are being assessed, and based upon the presented spectra I am perfectly satisfied that these superchiral field enhanced spectra are correct signals and not artefacts. While the samples used are still highly concentrated (5M) and the real test will come when studying low concentration spectra, the presented spectra are high quality and sufficient for supporting the authors' claims.

I think this is a very exciting report that will spur further development of chiroptical spectroscopies and smarter design of nanostructured surfaces for enhancing optical phenomena. I am very happy to support publication of this paper in Nature Communications.

Authors' response:

We thank the Reviewer for the positive comment and recognition of our work.

REVIEWER COMMENTS

Reviewer #1 (Remarks to the Author):

The author has responded fully to my comments and concerns.

Reviewer #2 (Remarks to the Author):

Dear editor and co-authors,

I read the response to all my concerns carefully. Although the authors addressed all of my questions, I'm not fully convinced by the answers to comments 1-4. Let me comment on them one by one.

Response to comment nr. 2:

I'm not convinced by the new words the authors use to describe what previously they called anapoles. In principle, toroidal and dipolar current distributions in a sphere should be orthogonal. Orthogonal current distributions cannot interact with each other and thus, strictly speaking, they do not hybridize. If anything, outside the sphere, the fields scattered by the toroidal moment and the dipolar moment interfere, but the modes do not hybridize. For a full and rigorous discussion on these topics, please see:

Nanz, Stefan. Toroidal multipole moments in classical electrodynamics: an analysis of their emergence and physical significance. Springer, 2016.

Response to comment nr. 3:

The authors did a good job at analyzing the temperatures that will be generated around the SI nanodiscs in comparison with metallic counterparts. Nevertheless, incidentally, they concluded that SI nanodiscs would induce temperature increments of 200°C. Although these temperatures are lower than those induced around plasmonic nanoparticles, I would not say that 200°C are low temperatures. Is there any report ensuring that such temperatures would not induce changes in the ROA spectra of the molecules under study?

Response to comment nr. 4:

Although now the theoretical analysis of the enhancement mechanisms is more complete, I'm not convinced by one approximation in the derivation. When the authors say

"while the optical chirality C^r of radiation at the wavelength of Raman scattering is significantly decayed and is approximately equal to zero in the near-field region (Supplementary Fig. 1)."

Following the derivation and looking at Supplementary Figure 1 I don't see any reason to believe that C^r can be neglected. Thus, I don't think that their conclusions on the theoretical model are well supported.

Also, in the line after Eq. 21, I think there is a typo. When they say R^e I think they mean R^r .

Last but not least, I cannot understand the paragraph between Eq.s 23 and 24 and it is an important step in the derivation.

In conclusion, this entire section needs to be revisited with care.

Additional note:

The authors overuse the term "superchiral fields". It is extensively known and proved that there exist no field distributions which are more chiral than circularly polarized light. What people call superchirality, is no other thing than enhanced chiral electromagnetic fields. In an ideal scenario, these fields are circularly polarized and enhanced in intensity. Although this "superchiral" term is somewhat standardized in the field, that does not mean it is correct, and in my opinion it should not be used.

In summary, the authors did a very big effort in addressing my concerns. Nevertheless, I'm not fully convinced by their reply. No matter what I still find the article relevant and interesting. I would be happy to review their article again. If they are able to provide a convincing answer to my queries, I would support the publication of the article in Nat. Comm.

To Reviewer #2

We are grateful to the Reviewer for taking the time to review our manuscript and give us his/her valuable comments. We have taken all the comments into consideration and have made appropriate changes to the manuscript. Our point-by-point response appears below, in which we first echo the Reviewer's comments (shown in italic) and then respond to them. All the changes to the manuscript and Supplementary Information are highlighted in red.

Reviewer #2's comment #1:

I'm not convinced by the new words the authors use to describe what previously they called anapoles. In principle, toroidal and dipolar current distributions in a sphere should be orthogonal. Orthogonal current distributions cannot interact with each other and thus, strictly speaking, they do not hybridize. If anything, outside the sphere, the fields scattered by the toroidal moment and the dipolar moment interfere, but the modes do not hybridize. For a full and rigorous discussion on these topics, please see: Nanz, Stefan. Toroidal multipole moments in classical electrodynamics: an analysis of their emergence and physical significance. Springer, 2016.

Authors' response:

We thank the Reviewer for the comment. We agree that the toroidal dipole and electric dipole modes are orthogonal and not coupled, but we do not think that the hybrid mode cannot be formed by a combination of two orthogonal modes. For example, the hybrid mode of a waveguide is a combination of two orthogonal modes, namely, transverse magnetic (TM) mode and transverse electric (TE) mode [Chen W. K., The Electrical Engineering Handbook, Elsevier, 2004]. Another example is *sp* hybridization in chemistry. The hybrid atomic orbital is also a combination of two orthogonal orbitals, namely, *s* orbital and *p* orbital in an atom [Wade L. G., Organic Chemistry, Pearson, (2012)]. Furthermore, it is important to note that the coupling between two modes is not essential for generating a hybrid mode. In general, a hybrid mode is a combination of several modes regardless of whether they are coupled or not. Therefore, we think it is reasonable to call the excited mode a hybrid mode. At the same time, we agree with the Reviewer that our previous statement "the hybrid mode is a

hybridization of electric and toroidal dipoles that results in their partial destructive interference in the far field” may be misleading because it may sound as if the electric dipole and toroidal dipole were coupled. To address the Reviewer’s comment, we have changed the name to a “dark” mode throughout the manuscript due to its small far-field radiation resulting from the destructive interference of the two modes. We hope the Reviewer is happy with this term.

Reviewer #2’s comment #2:

The authors did a good job at analyzing the temperatures that will be generated around the SI nanodiscs in comparison with metallic counterparts. Nevertheless, incidentally, they concluded that Si nanodiscs would induce temperature increments of 200°C. Although these temperatures are lower than those induced around plasmonic nanoparticles, I would not say that 200°C are low temperatures. Is there any report ensuring that such temperatures would not induce changes in the ROA spectra of the molecules under study?

Authors’ response:

We thank the Reviewer for the comment. To the best of our knowledge, we are not aware of any references that report the influence of such a temperature on ROA spectra. However, it can be inferred from our experimental results that the temperature did not induce any significant changes to ROA spectra as our measured ROA spectra agree well with those reported in previous literature [Hug, W. Virtual enantiomers as the solution of optical activity's deterministic offset problem. *Appl. Spectrosc.* 57, 1-13 (2003); Yamamoto, S. et al. Incident circularly polarized Raman optical activity spectrometer based on circularity conversion method. *J. Raman Spectrosc.* 41, 1664-1669 (2010)].

Reviewer #2’s comment #3:

Although now the theoretical analysis of the enhancement mechanisms is more complete, I'm not convinced by one approximation in the derivation. When the authors say "while the optical chirality C^r of radiation at the

wavelength of Raman scattering is significantly decayed and is approximately equal to zero in the near-field region (Supplementary Fig. 1)." Following the derivation and looking at Supplementary Figure 1 I don't see any reason to believe that C^r can be neglected. Thus, I don't think that their conclusions on the theoretical model are well supported. Also, in the line after Eq. 21, I think there is a typo. When they say R^e I think they mean R^r . Last but not least, I cannot understand the paragraph between Eq.s 23 and 24 and it is an important step in the derivation. In conclusion, this entire section needs to be revisited with care.

Authors' response:

We thank the Reviewer for the comment. We have revisited this section with much care. Here we want to clarify our analytical derivations as shown below.

As the approximation mentioned by the Reviewer is after Eq. (27), let us make the analytical derivation from Eq. (27) more detailed. Eq. (27) is given by

$$R_+^e R_+^r - R_-^e R_-^r \propto \left[\frac{\omega^e \widehat{\alpha}^e |\widetilde{\mathbf{E}}^e|^2}{2} + \frac{2C^e \widehat{G}^e}{\varepsilon_0} \right] \left[\frac{\omega^r \widehat{\alpha}^r |\widetilde{\mathbf{E}}^r|^2}{2} + \frac{2C_+^r \widehat{G}^r}{\varepsilon_0} \right] - \left[\frac{\omega^e \widehat{\alpha}^e |\widetilde{\mathbf{E}}^e|^2}{2} - \frac{2C^e \widehat{G}^e}{\varepsilon_0} \right] \left[\frac{\omega^r \widehat{\alpha}^r |\widetilde{\mathbf{E}}^r|^2}{2} + \frac{2C_-^r \widehat{G}^r}{\varepsilon_0} \right]. \quad (27)$$

If we expand the right-hand side of the equation, it can be expressed as

$$R_+^e R_+^r - R_-^e R_-^r \propto 2 \frac{2C^e \widehat{G}^e}{\varepsilon_0} \frac{\omega^r \widehat{\alpha}^r |\widetilde{\mathbf{E}}^r|^2}{2} + \frac{\omega^e \widehat{\alpha}^e |\widetilde{\mathbf{E}}^e|^2}{2} \left[\frac{2C_+^r \widehat{G}^r}{\varepsilon_0} - \frac{2C_-^r \widehat{G}^r}{\varepsilon_0} \right] + \frac{2C^e \widehat{G}^e}{\varepsilon_0} \left[\frac{2C_+^r \widehat{G}^r}{\varepsilon_0} + \frac{2C_-^r \widehat{G}^r}{\varepsilon_0} \right]. \quad (28)$$

This equation can be approximated as

$$R_+^e R_+^r - R_-^e R_-^r \propto 2 \frac{2C^e \widehat{G}^e}{\varepsilon_0} \frac{\omega^r \widehat{\alpha}^r |\widetilde{\mathbf{E}}^r|^2}{2} + \frac{\omega^e \widehat{\alpha}^e |\widetilde{\mathbf{E}}^e|^2}{2} \left[\frac{2C_+^r \widehat{G}^r}{\varepsilon_0} - \frac{2C_-^r \widehat{G}^r}{\varepsilon_0} \right], \quad (29)$$

where we have used $\frac{2C\widehat{G}}{\varepsilon_0} \ll \frac{\omega\widehat{\alpha}|\widetilde{\mathbf{E}}|^2}{2}$. To verify that this condition for the approximation is valid, we start with Eq.

(15), which relates C with $\widetilde{\mathbf{E}}$. For circularly polarized light, we obtain from Eq. (15)

$$C = \frac{\omega\varepsilon_0|\widetilde{\mathbf{E}}|^2}{2c}. \quad (30)$$

In addition, we have $|\widetilde{\mathbf{B}}| = \frac{|\widetilde{\mathbf{E}}|}{c}$ and $\widehat{G}|\widetilde{\mathbf{B}}| \ll \widehat{\alpha}|\widetilde{\mathbf{E}}|$ from Eq. (6) as $\widehat{G}|\widetilde{\mathbf{B}}|$ is a high-order term. Thus, we have

$\widehat{G} \ll c\widehat{\alpha}$. By simultaneously multiplying both the right- and left-hand sides of this relation by $\frac{2C}{\varepsilon_0}$ (where C is

positive based on our definition), we have

$$\frac{2C\hat{G}}{\varepsilon_0} \ll \frac{2Cc\hat{\alpha}}{\varepsilon_0}. \quad (31)$$

By substituting Eq. (30) into the right-hand side of Eq. (31), we have

$$\frac{2C\hat{G}}{\varepsilon_0} \ll \frac{\omega\hat{\alpha}|\hat{E}|^2}{2}. \quad (32)$$

Therefore, with this condition, $\frac{2C^e\hat{G}^e}{\varepsilon_0} \left[\frac{2C_+^r\hat{G}^r}{\varepsilon_0} + \frac{2C_-^r\hat{G}^r}{\varepsilon_0} \right]$ in Eq. (28) is a high-order term and can be omitted.

The first and second terms on the right-hand side of Eq. (29) correspond to incident circular polarization ROA (ICP-ROA) and scattered circular polarization ROA (SCP-ROA), respectively. For ICP-ROA, the optical chirality (circular polarization) of incident light (excitation field) C^e is well controlled while the optical chirality of Raman scattering (radiation field) C^r needs to be scrambled [Nafie L. A., *Vibrational optical activity: principles and applications*, Wiley, 2011]. This means that the circularly polarized components of Raman

scattering was eliminated in our measurement, which corresponds to $\frac{\omega^e\hat{\alpha}^e|\hat{E}^e|^2}{2} \left[\frac{2C_+^r\hat{G}^r}{\varepsilon_0} - \frac{2C_-^r\hat{G}^r}{\varepsilon_0} \right] = 0$. It is noted that, for all ICP-ROA measurements, one measures the scattered Raman intensities with either no polarization or some specified state of linear polarization without any circular or elliptical polarization content [Nafie L. A., *Vibrational optical activity: principles and applications*, Wiley, 2011]. Moreover, the silicon nanodisk array also scrambles the optical chirality of Raman scattering as it cannot enhance but even reduce the optical chirality from the near field to the far field at the Raman scattering wavelength (Supplementary Fig. 1) by considering optical reciprocity. For SCP-ROA, the optical chirality (circular polarization) of incident light (excitation field) C^e needs to be scrambled while the optical chirality of Raman scattering (radiation field) C^r is measured [Nafie L. A., *Vibrational optical activity: principles and applications*, Wiley, 2011]. This means the circularly polarized components of incident light was eliminated in the measurement, which corresponds to $2\frac{2C^e\hat{G}^e}{\varepsilon_0} \frac{\omega^r\hat{\alpha}^r|\hat{E}^r|^2}{2} = 0$.

As we used ICP-ROA in our experiment, $\frac{\omega^e\hat{\alpha}^e|\hat{E}^e|^2}{2} \left[\frac{2C_+^r\hat{G}^r}{\varepsilon_0} - \frac{2C_-^r\hat{G}^r}{\varepsilon_0} \right]$ in Eq. (32) can be omitted in our measurement. The measured ICP-ROA intensity can be rewritten as

$$R_+^e R_+^r - R_-^e R_-^r \propto \frac{2\omega^r\hat{\alpha}^r\hat{G}^e C^e |\hat{E}^r|^2}{\varepsilon_0}, \quad (33)$$

which is identical to our previous derivation where C^r is neglected. Therefore, we think the approximation is

reasonable for our ICP-ROA measurement.

Supplementary Fig. 1 | **a**, Average optical activity (RCP excitation) in the near-field region on the silicon nanodisk array as a function of wavelength (Raman shift), normalized by the optical chirality of incident CPL in the far field. **b**, Averaged square of the electric-field magnitude in the near-field region on the silicon nanodisk array as a function of wavelength (Raman shift), normalized by the square of the electric-field magnitude of incident CPL in the far field.

Regarding the typo in the line after Eq. (21) in our previous SI, we thank the Reviewer for pointing it out. We have corrected it in our revised SI.

As for the paragraph between Eq. (23) and Eq. (24) in our previous SI, to address the Reviewer's comment, we have revised it and made it more understandable in our revised SI. Since Raman scattering is a two-step process, we need to define the radiation step based on the excitation step. For the excitation rate of R_+^e , we define C_+^r as the optical chirality of radiation and define $\frac{1}{3}\widetilde{\mathbf{E}}_+^{r*} \cdot \widehat{\mathbf{A}}_{\gamma+}^r \nabla \widehat{\mathbf{E}}_{\gamma+}^r = T_+$ as the radiation contribution from the electric dipole-electric quadrupole polarizability. For the excitation rate of R_-^e , we define C_-^r as the optical chirality of radiation and define $\frac{1}{3}\widetilde{\mathbf{E}}_-^{r*} \cdot \widehat{\mathbf{A}}_{\gamma-}^r \nabla \widehat{\mathbf{E}}_{\gamma-}^r = T_-$ as the radiation contribution from the electric dipole-electric quadrupole polarizability.

In summary, we have revisited this section with great care. Please note that these analytical derivations are for readers to understand the underlying physics of enhanced ROA in the near field by conducting an order-of-magnitude estimation of enhancement factors, such that some approximations are used for simplification. We also want to emphasize that even with these approximations, the estimated order of magnitude of the enhancement factor agrees well with our experimental result.

Reviewer #2's comment #4:

The authors overuse the term "superchiral fields". It is extensively known and proved that there exist no field distributions which are more chiral than circularly polarized light. What people call superchirality, is no other thing than enhanced chiral electromagnetic fields. In an ideal scenario, these fields are circularly polarized and enhanced in intensity. Although this "superchiral" term is somewhat standardized in the field, that does not mean it is correct, and in my opinion it should not be used.

Authors' response:

We thank the Reviewer for the comment. We agree that "superchiral" is somewhat overused although it is standardized in the field. To address the Reviewer's comment, we have deleted "super" throughout the manuscript and SI.

Reviewer #2's comment #5:

In summary, the authors did a very big effort in addressing my concerns. Nevertheless, I'm not fully convinced by their reply. No matter what I still find the article relevant and interesting. I would be happy to review their article again. If they are able to provide a convincing answer to my queries, I would support the publication of the article in Nat. Comm.

Authors' response:

We thank the Reviewer for the comment and encouragement. We humbly hope that he/she is happy with our point-by-point response to his/her comments.

REVIEWERS' COMMENTS

Reviewer #2 (Remarks to the Author):

Dear editor and co-authors,

The authors addressed my last comments with care. I think the article is ready to be accepted.

Best wishes,

Aitzol Garcia-Etxarri

To Reviewer #2

We are grateful to the Reviewer for taking the time to review our manuscript and give us his valuable comment. Our point-by-point response appears below, in which we first echo the Reviewer's comments (shown in italic) and then respond to them.

Reviewer #2's comment #1:

Dear editor and co-authors,

The authors addressed my last comments with care. I think the article is ready to be accepted.

Best wishes,

Aitzol Garcia-Etxarri

Authors' response:

We thank the Reviewer for the positive comment and recognition of our work.